# Acceptability of COVID-19 self-testing among social and clinical vulnerable populations using a decentralized testing model in Abuja, Nigeria; A mixed methods analysis of an implementation study

Elvis Efe Isere[1], John Samson Bimba[1], Yasmin Dunkley[2]*, David Atuwo[1], Emily Nightingale[2], James Ekwu[1], Ambi Mamman Ibrahim[1], Gabriella Ofeh Adamu[1], Godpower Omoregie[3], Yamen Okonkwo[3], Nicola Desmond[4], Karin Hatzold[5], Elizabeth L. Corbett[2]

1 Zankli Research Centre, Bingham University, Karu, Nigeria, 2 Department of Clinical Research, Faculty of Infectious and Tropical Diseases, London School of Hygiene and Tropical Medicine, London, United Kingdom, 3 Society for Family Health, Abuja, Nigeria, 4 Department of Global Health and Development, Faculty of Public Health and Policy, London School of Hygiene and Tropical Medicine, London, United Kingdom, 5 Population Services International, Cape Town, South Africa

* yasmindunkley@hotmail.co.uk, yasmin.dunkley@lshtm.ac.uk

## Abstract

Diagnostic testing is critical during infectious disease outbreaks, enabling timely patient management and isolation to reduce transmission and mortality. During the COVID-19 outbreak in Nigeria, testing rates remained low due to limited access to centralized RT-PCR sites. To expand access, the National COVID-19 Testing Strategy (January 2021) introduced decentralized self-testing models targeting vulnerable populations. This study assessed the uptake of decentralized COVID-19 testing and the acceptability of self-testing among socially and clinically vulnerable populations in Abuja, Nigeria. A mixed-methods study was conducted across four primary health centres (PHC), four community pharmacies (CP), and four patent medicine stores (PMS) between October 2022 and May 2023. Symptomatic individuals received provider-delivered testing at PHC or provider-delivered/self-testing at CP and PMS using antigen rapid diagnostic tests (Ag-RDT). Social vulnerability was defined by low education, illiteracy, or low wealth; clinical vulnerability by age ≥ 50, unvaccinated status, or comorbidities. Testing uptake and acceptability were analyzed using logistic regression, while in-depth interviews (IDI) explored preferences for testing sites and methods. Of 1,586 individuals screened, 1,368 were eligible and 1,322 (96.6%) accepted testing. Most tests occurred at PHC (53.5%), followed by PMS (25.9%) and CP (20.7%). Social vulnerability was higher among PMS users than PHC users (OR = 1.37; 95% CI 1.05–1.77), while clinical vulnerability was lower at CP (OR = 0.24; 95% CI 0.16–0.35) and PMS (OR = 0.28; 95% CI 0.19–0.39) compared to PHC.

**Data availability statement:** All data can be found in the paper and Supporting information files.

**Funding:** This work was supported by Unitaid grant (2017-16-PSI-STAR to KH). The funders had no role in study design, data collection and analysis, decision to publish, or preparation of the manuscript.

**Competing interests:** The authors have declared that no competing interests exist.

Self-testing acceptability was high (93.4% at CP; 92.1% at PMS). Outcome of IDI highlighted trust in CP/PMS providers, proximity, convenience, and affordability as key drivers of testing uptake, with self-testing widely preferred across vulnerability groups. Decentralized testing through CP and PMS reached more socially vulnerable individuals and demonstrated high self-testing acceptability. Leveraging these outlets in outbreak responses could enhance equitable access to diagnostic testing in future pandemics.

## Introduction

The COVID-19 pandemic was declared a Public Health Emergency of International Concern by the World Health Organization (WHO) in January 2020 [1–3]. Diagnostic testing can play a critical role in managing outbreaks like COVID-19 through enabling patient management and isolating positive cases to reduce transmission, and associated mortality [4–10]. In Nigeria, expanding access to diagnostic testing was a key strategy to mitigate the COVID-19 pandemic's impact [4–10]. However, testing rates remained low partially due to limited access to centralized sites conducting RT-PCR. To address this, the Nigeria CDC introduced antigen-based rapid diagnostic tests (Ag-RDT) in November 2020, and later developed a National COVID-19 testing Strategy in January 2021 to explore decentralized self-testing models [4–10].

Centralized testing models disproportionately affect socially vulnerable groups, including those living in rural areas, individuals with limited financial means, and people with underlying health conditions, who often face additional barriers to accessing care [11]. Rural communities throughout Nigeria routinely face greater healthcare disparities, given remote geographies, poorer infrastructure, and higher poverty rates which can limit access to testing and treatment [12]. The economic shocks of COVID-19 amplified vulnerabilities: lockdowns and inflated costs fell hardest on low-income and informal sector families, deepening exclusion from centralized services [13–14]. Existing inequities heighten the clinical risk of undiagnosed disease among these vulnerable populations.

Decentralized testing models explored in Nigeria included distribution of self-tests through Community Pharmacies (CP) and Patent Medicine Stores (PMS); these offer a promising solution to the limitations of centralized testing facilities. Global evidence from decentralized HIV testing service delivery models have demonstrated reductions in structural barriers and expanded reach among underserved populations [15,16]. Decentralizing testing sites and technologies can increase accessibility, through reducing structural barriers such as travel distance, waiting times, and stigma for vulnerable populations who may otherwise not access services [11].

In Nigeria, PMS and CP have increasingly been leveraged to deliver essential health services in vulnerable communities particularly in areas with limited access to formal healthcare [17–29]. PMS have been documented as the first point of care for many Nigerians, especially in rural areas, providing treatment for malaria, diarrhea, pneumonia, and family planning services for both children and adults. Decentralized

malaria rapid diagnostic tests and treatment initiations in CP and PMS have improved malaria case detection and treatment accuracy in Lagos [28]. CP have also been explored as delivery models for HIV pre-exposure prophylaxis (PrEP) and are recognized for their role in integrating family planning with HIV services in resource-limited urban settings [30].

Given the potential of decentralized models to increase access for vulnerable populations against the backdrop of pandemic, this study aims to evaluate the uptake of decentralized COVID-19 testing models, and the acceptability of decentralized COVID-19 self-testing models among socially and clinically vulnerable populations in Abuja. The findings aim to inform public health strategies to enhance testing accessibility and equity, ensuring vulnerable populations are well integrated in the fight against pandemic-potential diseases.

## Methods

### Study design

A mixed method study used cross-sectional analysis following the Strengthening the Reporting of Observational Studies in Epidemiology (STROBE) guidelines [31] and in-depth interviews (IDI) was employed in the study among symptomatic clients presenting at CP, PMS and PHC in the 6 area councils of the Federal Capital Territory (**Fig 1**).

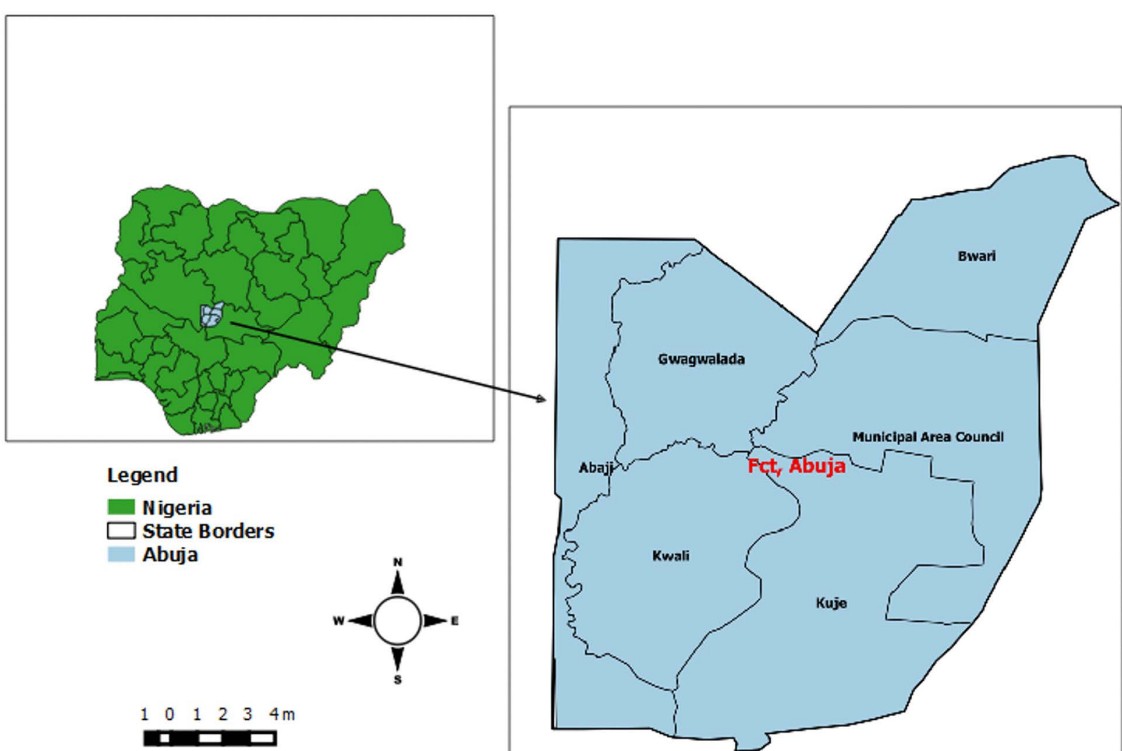

The map was created in QGIS using the shapefile from the Geo-Referenced Infrastructure and Demographic Data for Development (GRID3-NGA) database, available at the GRID3 data repository, under CC BY 4.0: https://data.grid3.org/datasets/GRID3::grid3-nga-operational-lga-boundaries/about

**Fig 1. Map of Nigeria and the Federal Capital Territory showing the six area councils.** The map was created in QGIS using the shapefile from the Geo-Referenced Infrastructure and Demographic Data for Development (GRID3-NGA) database, available at the GRID3 data repository, under CC BY 4.0 License: https://data.grid3.org/datasets/GRID3:grid3-nga-operational-lga-boundaries/about.

Clients visiting designated study locations were screened for eligibility regardless of services sought. Eligibility was determined as being 18 years or older and exhibiting at least one COVID-19 symptom (S1 File). The inclusion of these symptoms as eligibility criteria for this study was based on their documented association with COVID-19 across multiple clinical and epidemiological studies [32–36]. There were no literacy or language eligibility requirements.

Eligible individuals were enrolled into the study from 19th October 2022–30th May 2023. Study locations comprised four CP, four PMS, and four PHC across the six area councils of the Federal Capital Territory. Participants who screened positive for COVID-19 symptoms were offered provider-delivered testing at PHC or a choice between provider-delivered and self-testing using the Standard Q COVID-19 Ag Home test kit (SD Biosensor, Republic of Korea) at CP and PMS. For those who opted for self-testing, follow-up calls were made on the third day post-test to obtain results and outcomes.

Participants testing positive through self-testing were referred for confirmatory testing using PCR or Ag-RDT at a health facility. Participants were also linked with the Federal Capital Territory COVID-19 case management team for follow-up, including contact tracing. A subset of participants were purposively sub-sampled and engaged for qualitative IDI following testing, based on age, gender, educational status and testing facility (**Fig 2**).

Data was collected through structured quantitative survey-tools and semi-structured qualitative interviews. A participant log and structured questionnaire were used to obtain demographic data, including sex, age, and education level, along with socioeconomic indicators such as food security status and literacy. Socioeconomic indicators were self-reported by participants using questions from previously validated tools (the Household Food Insecurity Access Scale (HFIAS) and national literacy assessment items) [37]. Information on COVID-19 symptoms, perceived severity, risk perception, vaccination status, prior hospitalization due to COVID-19, testing preferences (self-testing vs. provider-administered testing), and testing outcomes were also collected. Clinical history was self-reported by participants unless otherwise documented in accompanying referral slips or health cards. IDI were conducted among selected participants to gain detailed insights into their experiences with COVID-19 testing, preferences for specific test methods, and factors influencing their acceptance of self-testing. Surveys and interviews were conducted in English, Pidgin or Hausa, depending on participant preference. Trained multilingual interviewers facilitated this process.

## Variable selection and data analysis

The primary outcome was the difference in uptake of COVID-19 testing interventions – including self-testing - by facility, defined as the percentage of eligible clients who accepted COVID-19 Ag-RDT testing when offered across PHC, CP, and PMS settings. The secondary outcome was the acceptability of COVID-19 self-testing measured as the percentage of eligible clients who opted for self-testing over provider-delivered testing in PMS and CP.

Quantitative analysis was conducted in Stata/IC version 15.0 (StataCorp LLC). We described the overall numbers of clients accessing testing across all facilities, categorized by facility type (uptake) and summarized participants' characteristics both overall and by testing settings. Uptake and acceptability data were analyzed using selected variables: participants' demographics and clinical and social vulnerability. Demographic variables included age, which was categorized into two groups: participants under 50 years of age and those 50 years or older. Gender was recorded as male or female, and its distribution was examined across the different test settings (PHC, CP, and PMS). We hypothesized differences in social and clinical vulnerability among individuals in different study settings. Participants were classified as socially vulnerable if they reported less than secondary-level education, illiteracy, or self-assessed as poorest in wealth. Participants were classified as clinically vulnerable if they were 50 years or older, had one or more specified comorbidities (S2 File), or were unvaccinated. We examined associations between social and clinical vulnerability factors and primary outcomes at study settings using unadjusted logistic regression. Clinical symptoms are presented as supplementary analysis (S3 File).

IDI audio recordings were manually transcribed, and transcripts uploaded into NVivo Pro version 12.0. We conducted thematic content analysis of qualitative data, using multiple coding cycles to generate an inductive codebook. Constant comparative coding was used to identify areas of consensus and divergence on the topic of uptake of COVID-19 testing

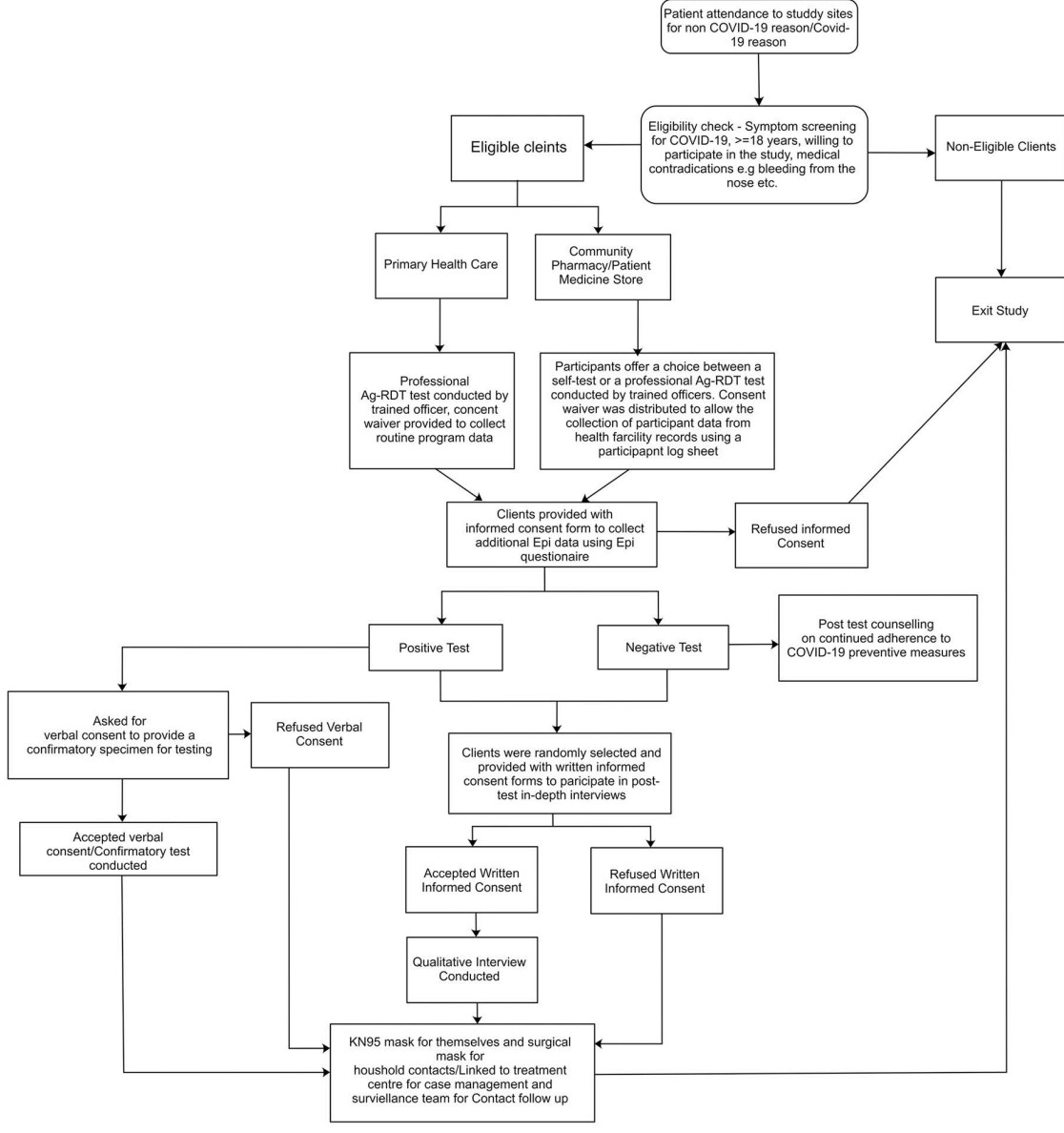

**Fig 2. Study procedure overview.**

by social and clinical vulnerability status, including between test settings, and acceptability of self-testing or provider testing until saturation was reached. Final themes were agreed upon by EEI and YD.

## Ethical considerations

Ethical approval for this study was obtained from the WHO Ethical Review Committee (Protocol ID: CERC.0165), the London School of Tropical Medicine and Hygiene Intervention Research Ethics Committee (Project ID: 26886), and the Federal Capital Territory Authority (FCTA), Abuja, Nigeria Ethical Review Committee (Approval number: FHREC/2022/01/29/09-03-22).

Consent waivers were granted to access routine health data from health facility records. Verbal consent was obtained for the collection of supplementary epidemiological data through questionnaires. Written informed consent was obtained for qualitative interviews.

## Results

Across all study settings, 1,586 hospital clients were screened for study eligibility, of which 1,368 (PHC; 718, PMS; 356, CP; 294) were eligible. Of these, 1,322 (96.6%) accepted COVID-19 Ag-RDT testing. Most participants tested through PHC (707/1,322, 53.5%), followed by PMS (342/1,322, 25.9%) and CP (273/1,322, 20.7%).

Most participants were aged below 50 years of age (**Table 1**): PHC (664/707, 93.9%), CP (254/273; 93.4%) and PMS (299/342, 87.4%).

Gender distribution varied across study settings; women made up less than half the participants in PMS (170/342, 49.7%), but almost two-thirds of participants in CP (167/273, 61.2%) and three-quarters in PHC (546/707, 77.2%). Educational levels differed as well, with more tertiary-educated participants in PMS (163/342, 47.7%) and CP (131/273, 48.0%) than in PHC (231/707, 32.7%). There were differences in self-assessed wealth, with most ranking their wealth in the middle category, but a higher proportion of participants from CP identified as poorest (37/273, 13.6%) compared to PHC (20/707, 2.8%) and PMS (3/342, 0.9%).

**Table 1. Participants' demographics profile by study setting.**

| Variable | Total N = 1322 | PHC N = 707 | CP N = 273 | PMS N = 342 |
|---|---|---|---|---|
| **Demographic characteristics** | | | | |
| **Age Group** | | | | |
| <50 years | 1217(92.1%) | 664(93.9%) | 254(93.0%) | 299(87.4%) |
| ≥50 years | 105(7.9%) | 43(6.1%) | 19(7.0%) | 43(12.6%) |
| **Sex** | | | | |
| Female | 883(66.8%) | 546(77.2%) | 167(61.2%) | 170(49.7%) |
| Male | 439(33.2%) | 161(22.8%) | 106(38.8%) | 172(50.3%) |
| **Education level** | | | | |
| None | 25(1.8%) | 18(2.6%) | 1(0.4%) | 6(1.8%) |
| Primary | 92(7.0) | 58(8.2%) | 3(1.1%) | 31(9.1%) |
| Secondary | 667(50.4%) | 399(56.4%) | 128(46.9%) | 140(40.9%) |
| Tertiary | 526(39.8%) | 231(32.7%) | 131(48.0%) | 163(47.7%) |
| Others | 13(1.0%) | 1(0.1%) | 10(3.7%) | 2(0.6%) |
| **Ability to read one page of English/local language (Literacy)** | | | | |
| Yes | 1243 (94.0%) | 661(93.5%) | 264(96.7%) | 318(93.0%) |
| No | 79(6.0%) | 46(6.5%) | 9(3.3%) | 24(7.0%) |
| **Self-assessed wealth** | | | | |
| Poorest | 60 (4.5%) | 20 (2.8%) | 37 (13.6%) | 3 (0.9%) |
| Middle | 1238 (93.6) | 676 (95.6) | 226 (82.7) | 336 (98.2) |
| Richest | 22 (1.7%) | 9 (1.3%) | 10 (3.7%) | 3 (0.9%) |
| **Perceived Household Feeding Concerns*** | | | | |
| Yes | 392 (29.7) | 193 (27.3) | 71 (26.0) | 128 (37.4) |
| No | 928 (70.2) | 514 (72.7) | 201(73.6) | 213 (62.3) |

* Perceived Household Feeding Concerns refers to a situation where a participant reports being worried or anxious about their households ability to provide enough food for its members a week prior the study.

More than three quarters of participants (1003/1322; 75.9%) had no comorbidities, with PHC having the highest proportion of participants with comorbidities (253/707, 35.8%), compared to CP (27/273, 9.9%) and PMS (39/342, 11.4%) (**Table 2**).

Socially vulnerable participants made up 267/707 (37.8%) in PHC 109/273 (39.9%) in CP, and 155/342 (45.3%) in PMS, indicating a higher percentage of social vulnerability in PMS (p=0.064). Clinically vulnerable participants were predominantly in PHC (296/707, 41.9%), while CP (40/273, 14.7%) and PMS (57/342, 16.7%) had significantly lower rates, highlighting a notable difference in clinical vulnerability across the settings.

Compared with their PHC counterparts, participants in the PMS group had 37% greater odds of being socially vulnerable (OR: 1.37, 95% CI 1.05 to 1.77; p=0.019). There was no increase in the odds of social vulnerability for CP (OR: 1.11, 95% CI 0.84 to 1.48; p=0.466). When clinical indicators were combined with severe COVID-19 outcomes, participants had 76% and 72% lower odds of being clinically vulnerable in CP (OR: 0.24, 95% CI 0.16 to 0.25, p<0.001) and PMS (OR: 0.28, 95% CI 0.19 to 0.39, p<0.001), respectively, than in PHC (**Table 3**).

However, despite this increased clinical vulnerability, test positivity rates per 100 tests were low across all study settings; PHC showed a positivity rate of 10/706 (1.4%), CP at 3/273 (1.1%), and PMS at 2/342 (0.6%) (p=0.682).

Finally, almost all study participants in CP (255/273, 93.4%) and PMS (315/342, 92.1%) opted for self-testing, indicating a high acceptability of self-testing for COVID-19 among study participants (p<0.001).

**Table 2. Clinical profile and COVID-19 Self perceived risk of participants by study setting.**

| Variable | Total N=1323 | PHC N=707 | CP N=273 | PMS N=342 |
|---|---|---|---|---|
| **Comorbidities** | | | | |
| None | 1004(75.8%) | 454(64.2%) | 246(90.1%) | 303(88.6%) |
| One comorbidity | 171(12.9%) | 131(18.5%) | 19(7.0%) | 21(6.1%) |
| Two comorbidities | 170 (12.8%) | 69 (9.8%) | 7 (2.6%) | 18 (5.3%) |
| Three or more comorbidities | 201 (15.2%) | 53 (7.5%) | 1 (0.4%) | 9 (2.6%) |
| **Self-perceived severity of symptoms** | | | | |
| Mild | 1121(84.8%) | 639(90.4%) | 223(81.7%) | 259(75.7%) |
| Severe | 94(7.1%) | 38(5.4%) | 18(6.6%) | 38(11.1%) |
| **Self-perceived risk of severe COVID-19 outcome** | | | | |
| Unlikely | 732 (55.4) | 495 (70.0) | 104 (38.1) | 133 (38.9) |
| Neutral | 258 (19.5) | 91 (12.9) | 131 (48.0) | 36 (10.5) |
| Likely | 322 (24.4) | 119 (16.8) | 31 (11.4) | 172 (50.3) |
| Unsure | 7 (0.5) | 0 | 7 (2.6) | 0 |
| **Vaccination Status** | | | | |
| Unvaccinated | 1095 (82.8) | 561 (79.4) | 218 (79.8) | 316 (92.4) |
| Vaccinated | 227 (17.2) | 146 (20.6) | 55 (20.2) | 26 (7.6) |
| **Days since onset of symptoms\*** | | | | |
| 1–4 days | 1020(77.2%) | 502(71.0%) | 252(92.3%) | 266(77.8%) |
| 5 days and above | 300(22.7%) | 203(28.7%) | 21(7.7%) | 76(22.2%) |
| **Ever previously tested for COVID-19** | | | | |
| No | 1069 (80.9%) | 586(83.0%) | 227(83.2%) | 256(74.9%) |
| Yes | 252(19.1%) | 120(17.0%) | 46(16.9%) | 86(25.2%) |
| **Know someone who has died from COVID-19** | | | | |
| No | 1277(96.8%) | 669(94.9%) | 267(97.8%) | 341(100.0%) |
| Yes | 42(3.2%) | 36(5.1%) | 6(2.2%) | 0(0.0%) |

**Table 3. Logistic regression analysis of social and clinical vulnerability in testing uptake by study setting.**

| Setting | Socially vulnerable | | | Clinically vulnerable | | |
|---------|---------------------|--|--|-----------------------|--|--|
| | **Unadjusted OR** | **p value** | **95% CI** | **Unadjusted OR** | **p value** | **95% CI** |
| PHC (base) | | | | | | |
| CP | 1.09528 | 0.533 | 0.823 - 1.458 | 0.23837 | <0.001 | 0.163 - 0.349 |
| PMS | 1.36594 | 0.0194 | 1.051 - 1.776 | 0.2777 | <0.001 | 0.199 - 0.387 |

### In-depth interview outcomes

22 of 45 participants approached for IDI agreed to participate: PHC (8/22, 36.3%), CP (7/22, 31.8%) and PMS (7/22, 31.8%). Demographics are described in S4 File.

Uptake in decentralized facilities was shaped by multiple factors, particularly trust in decentralized healthcare providers and perceived accessibility – especially for those with social and clinical vulnerability. Self-testing represented a further extension of accessibility – with no differences identified for social or clinically vulnerable participants.

### Factors influencing uptake of decentralized COVID-19 testing services

Trust played a fundamental role in individuals' willingness to accept testing at PMS. Medicine sellers served as trusted healthcare figures in the community with their recommendations significantly influencing health-behaviors. Many participants noted that they had long relied on their local medicine sellers for treatment and medications, fostering confidence in their recommendations including for COVID-19 testing:

*"Since I was sick and the medicine seller informed me about the COVID-19 test, and I have confidence and trust in whatever drug he sold to me for malaria treatment before now, I decided to follow his instruction and do the COVID-19 test he recommended to me."* (**PMS, Female, 24 years**)

Participants also highlighted the advantage of avoiding long trips to distant hospitals, where testing procedures often require extended waiting times. One respondent expressed frustration with the hospital process:

*"If it was in the hospital, the stress will be too much. The process before you see the lab officials is too long; you will be on the queue for a long time. But here in the PMS, you may not be able to meet many people, and they will attend to you in time so that you will not waste your whole schedule."* (**PMS, Female, 40 years**)

This was particularly pronounced amongst socially vulnerable individuals with financial constraints, where PMS were seen as accessible ways to get tested without the logistical and bureaucratic hurdles often associated with hospital visits. Resultingly, cost emerged as a critical factor influencing uptake. Many participants shared that they would have been unable to afford a COVID-19 test if it were not offered for free at PMS:

*"Most people within this and many surrounding communities couldn't afford frequent COVID-19 testing at hospitals, so they had to rely on places that were easier to access and the COVID-19 test done free of charge, like pharmacies and patent medicine stores."* (**PMS, Male, 37 years**)

Decentralized testing conducted at CP and PMS was also thought to benefit those with clinical vulnerability:

*"It is very important for those that have underlying illnesses such as diabetes like I do to access the COVID-19 test in pharmacies and ensure they do not have COVID-19, which can lead to death."* (**CP, Male, 72 years**)

## Self-testing acceptability

Most participants preferred self-tests over professionally delivered COVID-19 testing. The convenience of the decentralized testing site was further complemented by the convenience provided of a decentralized testing technology – able to be conducted within the home:

> "*Of course, the convenience is very good. I took the test kit home and did the test myself following the instructions, and I sent the test result to the PMS owner, for me doing the test at the convenience of your home is very good, you can test yourself as many times you want when you are sick at home and know when to quickly seek treatment if you are positive for COVID-19 than waiting to get to see a doctor or nurse before getting tested*" [**PMS, Male, 73 years**].

This was uniform across study participants with no differences noted by social or clinical vulnerability status.

## Discussion

Our study found high uptake amongst socially vulnerable populations for COVID-19 testing implemented through decentralized CP and PMS in Nigeria. We also found high acceptability for self-testing for COVID-19 at these sites. Although overall uptake of COVID-19 testing across CP and PMS was less than PHC over the same time-period, both the CP and PMS model reached a higher proportion of socially vulnerable individuals. A higher proportion of participants from CP identified as being in the poorest wealth category, potentially representative of the location of CP in more urban settings where income inequality may be starker than rural locations. PMS models also reached more men; a group generally underrepresented in health service provision. Although a higher proportion of clinically vulnerable clients utilized PHC outlets for testing, test positivity rates remained low throughout reflecting the implementation of this study during a period of low community transmission towards the end of the COVID-19 pandemic. Almost all study participants in CP and PMS opted for self-testing – including taking kits home – to report their results later. This indicated a high acceptability of self-testing for COVID-19 among study participants.

It was unsurprising that a higher proportion of socially vulnerable individuals accessed testing through the CP and PMS. Our qualitative findings reinforced existing research that CP and PMS can serve as trusted, accessible, and cost-effective alternatives to centralized-testing addressing key barriers such as affordability, and location [17,21,38–43]. The acceptability of COVID-19 self-testing observed in our research also aligns with findings from prior studies in low- and middle-income countries where comparable acceptability rates of COVID-19 self-testing were reported among diverse groups [44–46]. Our acceptability data also provides additional empirical evidence for a community survey conducted in Nigeria which demonstrated willingness (81.8%) among the Nigerian population to utilize COVID-19 self-test kits if they were available in the country [44].

The high uptake and acceptability of COVID-19 self-testing through CP and PMS models in Nigeria demonstrates a promising, scalable and equitable approach for improving access to testing for other conditions, including, potentially, pandemic-potential pathogens. Leveraging these community-based outlets, especially PMS, which effectively reaches socially vulnerable populations, can theoretically enhance early diagnosis, reduce healthcare access barriers, and support decentralized public health strategies. Self-testing technologies can further decentralize diagnostics into the home, aligning with expressed user preferences. More research is needed on the uptake and acceptability of these models and technologies in periods of acute disease transmission and for other emerging disease conditions. How this is taken forward and integrated into public health programming is a key next step.

This study is not without limitations; initially, the self-test kits utilized in our study provided instructions exclusively in English, potentially introducing selection bias given the aim to reach socially vulnerable populations. The instructions for use were later optimized for other languages and visually for those who were non-literate. This was also a study

conducted in one, albeit multi-ethnic, location (Abuja Federal Capital Territory). To inform scale-up, more research on the intervention context would be required to explore potential implementation barriers.

## Conclusion

Our study highlights the potential of decentralized testing models and self-testing as valuable tools for future outbreak responses in Nigeria. It offers the possibility of decentralizing testing beyond designated centralized health facilities, increasing access for vulnerable individuals at the community level - a promising delivery model for potential scaling-up in the context of future pandemic preparedness.

## Supporting information

**S1 File. Participant eligibility criteria for study inclusion.**
(DOCX)

**S2 File. List of comorbidities self-reported by study participants.**
(DOCX)

**S3 File. Clinical presentation.**
(DOC)

**S4 File. Demographics of IDI participants.**
(DOC)

**S1 Data. Data file for the presented analysis.**
(XLS)

**S1 Checklist. PLOS inclusivity in global research checklist.**
(DOC)

## Acknowledgments

We sincerely appreciate the Nigeria Centre for Disease Control and Prevention (NCDC), the Department of Public Health, FCTA, Abuja and the Primary Health Care Development Board, FCTA, Abuja, our implementing partner, Society for Family Health, Nigeria, Pharmaceutical Council of Nigeria, Association of Patent Medicine Vendors Abuja, the research assistants and supervisors, and all study participants.

## Author contributions

**Conceptualization:** Yasmin Dunkley, Emily Nightingale, Nicola Desmond, Karin Hatzold, Elizabeth L. Corbett.

**Data curation:** Elvis Efe Isere, John Samson Bimba, David Atuwo, Gabriella Ofeh Adamu.

**Formal analysis:** Elvis Efe Isere.

**Funding acquisition:** Karin Hatzold, Elizabeth L. Corbett.

**Investigation:** Elvis Efe Isere, Yamen Okonkwo.

**Methodology:** Elvis Efe Isere, Yasmin Dunkley, David Atuwo, Emily Nightingale, James Ekwu, Ambi Mamman Ibrahim, Gabriella Ofeh Adamu, Godpower Omoregie, Nicola Desmond, Elizabeth L. Corbett.

**Project administration:** Elvis Efe Isere, James Ekwu, Ambi Mamman Ibrahim, Godpower Omoregie, Yamen Okonkwo, Karin Hatzold.

**Resources:** John Samson Bimba, Godpower Omoregie, Karin Hatzold, Elizabeth L. Corbett.

**Supervision:** John Samson Bimba, Yasmin Dunkley, James Ekwu, Ambi Mamman Ibrahim, Godpower Omoregie, Yamen Okonkwo, Nicola Desmond, Karin Hatzold, Elizabeth L. Corbett.

**Writing – original draft:** Elvis Efe Isere.

**Writing – review & editing:** John Samson Bimba, Yasmin Dunkley, David Atuwo, Emily Nightingale, James Ekwu, Ambi Mamman Ibrahim, Gabriella Ofeh Adamu, Godpower Omoregie, Yamen Okonkwo, Nicola Desmond, Karin Hatzold, Elizabeth L. Corbett.

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
