## [Decision Letter · Decision Letter 0]

17 Jun 2025

PGPH-D-25-00883

Acceptability of COVID-19 Self-testing among Social and Clinical Vulnerable Populations using a Decentralized Testing Model in Abuja, Nigeria; A mixed methods analysis of an implementation study.

Dear Dr. Dunkley,

Thank you for submitting your manuscript to PLOS Global Public Health. After careful consideration, we feel that it has merit but does not fully meet PLOS Global Public Health’s publication criteria as it currently stands. Therefore, we invite you to submit a revised version of the manuscript that addresses the points raised during the review process.

Three reviewers have assessed the manuscript and provided their comments below. The requests focus on the need for refinement in the introduction, additional detail in the methodology such as including the selection criteria for participants and a more in-depth discussion on the significance of these findings for conditions other than COVID-19.

Please review their comments and make the appropriate revisions to address the concerns raised. 

We look forward to receiving your revised manuscript.

Kind regards,

Emma Campbell, Ph.D

Staff Editor

Journal Requirements:

2. In the online submission form, you indicated that [Data is available upon request from the authors.].

a. In a public repository,

b. Within the manuscript itself, or

c. Uploaded as supplementary information.

3. We do not publish any copyright or trademark symbols that usually accompany proprietary names, eg (R), (C), or TM (e.g. next to drug or reagent names). Please remove all instances of trademark/copyright symbols throughout the text, including ® and ™ on page 6.

4. Figure 1: please (a) provide a direct link to the base layer of the map (i.e., the country or region border shape) and ensure this is also included in the figure legend; and (b) provide a link to the terms of use / license information for the base layer image or shapefile. We cannot publish proprietary or copyrighted maps (e.g. Google Maps, Mapquest) and the terms of use for your map base layer must be compatible with our CC-BY 4.0 license.

Reviewers' comments:

Reviewer's Responses to Questions

**Comments to the Author**

1. Does this manuscript meet PLOS Global Public Health’s publication criteria?

Reviewer #1: Yes

Reviewer #2: Partly

Reviewer #3: Yes

2. Has the statistical analysis been performed appropriately and rigorously?

Reviewer #1: Yes

Reviewer #2: Yes

Reviewer #3: Yes

3. Have the authors made all data underlying the findings in their manuscript fully available (please refer to the Data Availability Statement at the start of the manuscript PDF file)?

Reviewer #1: Yes

Reviewer #2: Yes

Reviewer #3: Yes

4. Is the manuscript presented in an intelligible fashion and written in standard English?

Reviewer #1: Yes

Reviewer #2: Yes

Reviewer #3: Yes

Reviewer #1: This is a mixed-methods study exploring de-centralised and self-testing for COVID-19 in Abuja, Nigeria. The qualitative findings provide useful insights into the reasons why people prefer to access testing at different types of providers and into how testing can be expanded to those with limited access. The study would benefit by more detail on how participants were selected for the qualitative component and a more in-depth discussion on the significance of these findings for conditions other than COVID-19. More specific comments below.

Lines 70-71 “Diagnostic testing plays a critical role in managing outbreaks like COVID-19 by enabling patient management and isolating positive cases to reduce mortality” – please rephrase as the direct aim of isolation is not reducing mortality but transmission (reduced mortality is a later consequence of decreased transmission)

Line 75 “Nigeria CDC introduced antigen-based rapid diagnostic tests (Ag-RDT) in 2020” – specify in which month rapid testing was introduced

Line 117: describe in more detail in the methods how the random collection of participants for the interviews was conducted. Were the interviews conducted on the same day as the testing?

Table 2: explain what “Household feeding concern means” and the reason why the proportion of those with feeding concerns is quite different than of those in the poorest stratum of wealth (29.7% vs. 4.5%)

Line 184: it would be interesting to know why the percentages of women attending the different types of facilities were different and also discuss the differences in income and attendance to the different types of facilities.

Table 3: would the authors be able to expand the categories for the number of comorbidities with an additional stratum. This would be to better understand how many participants had multiple co-morbidities (while excluding older age and pregnancy)

Line 220 (or at the beginning of the results section) report on the numbers approached for IDI and on how many refused to be interviewed

Table 5: clarify whether the logistic regression analysis was adjusted or unadjusted (I assume unadjusted)

Lines 379-385: the sentence is very long, please break down to improve readability

Add some context to better understand the low positivity during the study period – was the study conducted during a nationwide Covid-19 “wave” or during a low-intensity transmission period?

Discuss in more detail how the findings from this study could be applied for scaling up and improving access to testing for other conditions (e.g future perspectives)

Revise spelling as there are some minor typographical errors (e.g. missing plurals)

Reviewer #2: This is great research to help demonstrate the use of self-testing for transmissible diseases and utilize a mixed methods model to understand the potential benefits and acceptability of these tests.

Introduction:

This introduction set the context up well for readers to understand the gaps and objectives of this project.

I would have liked to see more about how this study relates to other research out there around the acceptability of self-testing or perhaps other health outreach models that showed improvement when expanding to these more community-based clinic settings.

The final paragraph states that the testing at the CP and PMS will be compared to the provider delivered testing at the PHCs but I don’t see that concept again throughout the rest of the manuscript leading me to wonder what role the PHCs had in this study.

Methods:

There are a few areas in the methods that I do not feel were described in sufficient detail to be reproducible.

Please add rationale or a citation for why you selected the symptom requirements you did for eligibility.

Figure 2 should be redone. It is blurry and not readable.

I’d like to see more details on the data collection: were the socioeconomic indicators self-reported or did you use a validated questionnaire? Please describe in more detail and/or add a citation. It is also unclear to me what data were collected through medical records, and which were self-reported or through the participant log.

I’d also like to see more details on how participants were randomly selected for the interviews. Were there any considerations for balanced sex, age, location to ensure a balanced group. Was saturation determined?

Please describe in the methods if the surveys and interviews were conducted in English or other languages? The limitations state the instructions for the self-test were in English, please describe if language or literacy was also an eligibility requirement.

Results:

Quantitative results are straightforward.

For the interviews, please add how many were invited to participate in interviews and describe the demographics of the population that agreed/completed the interviews.

I appreciate the themes discussed in the qualitative results but do feel like sections are redundant and could be written in a more concise manner.

Miscellaneous:

I got the impression various sections were written by different authors and could use a review by the first author to ensure consistency in formatting and language. There are multiple instances of re-defining acronyms that have already been defined (e.g., “patent medicine stores (PMS)”) – define the first time then use PMS throughout. Also, one of the qualitative themes has the quotes in the paragraph where others have quotes as their own paragraph.

I think this paper could use more editing to be more concise, avoid redundancies, and run on sentences.

Reviewer #3: In their study “Acceptability of COVID-19 Self-testing among Social and Clinical Vulnerable Populations using a Decentralized Testing Model in Abuja, Nigeria; A mixed methods analysis of an implementation study” Isere et al. assess the acceptability of as well as the advantages and challenges when implementing COVID-19 self-testing through different modes of care. Overall, I find the study very well conducted and results laid out clearly. However, I have two comments concerning (1) the focus and novelty of results and (2) the length of the manuscript and abstract, which are further outlined below.

1) The overall focus of the manuscript is not yet clear to me. Based on its current version, I believe the authors concentrated on three questions: first, to assess the general uptake of Ag-RDT testing for SARS-CoV-2 (see line 134); second, to evaluate the general acceptability of SARS-CoV-2 self-testing (see line 135); third, to analyze differences in uptake and acceptability between centralized versus decentralized test distribution (i.e., between community pharmacies and patent medicine stores versus primary health centers). The first and second question have already been discussed widely in scientific literature. The third question, however, has not yet been evaluated thoroughly and answering it could provide novel perspectives on implementing antigen (self-)testing for SARS-CoV-2. Therefore, I would suggest to refine the manuscript’s introduction, presentation of results, and discussion, to focus only on evaluating the difference in testing uptake and acceptability between the different modes of care (of note, I don’t expect the authors to change the analysis underlying their manuscript or to make larger adjustments in the results section – I believe the analysis and results already entail all relevant aspects to assess differences in participation in SARS-CoV-2 antigen testing between different modes of test distribution). Specifically, the authors could adjust the title of the manuscript, e.g., along the lines of “Centralized versus decentralized test distribution models to enhance COVID-19 antigen testing participation in vulnerable populations: a mixed-methods study”. Furthermore, in the introduction, I would suggest removing line 71 (“In Nigeria, …”) to 77 (“… self-testing.”). Building on what is already stated in line 78 to 85, the freed space could be used to highlight clearly why vulnerable populations in Nigeria are disadvantaged by centralized modes of test distribution and how these disadvantages could be resolved through decentralized testing. In the methods, line 134-135, I would suggest to define the *difference* in uptake and acceptability as the primary and secondary outcome. In the results section and discussion, I would suggest to remove all text not specifically related to assessing the difference between centralized and decentralized testing (see my second comment).

2) To enhance readability, I would suggest to shorten the manuscript on several instances. First, I would ask to limit the abstract to a maximum of 350 words. Secondly, when presenting the main characteristics of the study population in lines 171 to 197, I would suggest to focus the text on those 5-6 characteristics of the study participants that are most essential for understanding the study’s conclusion. Thirdly, I would suggest to move the section on ‘Preference for self-test over professional delivered testing’ to the Supplements, aside from those parts of this section that specifically add to understanding the impact of centralized versus decentralized test distribution models (see my first comment). Similarly, in the discussion, I would suggest to remove lines 343-370 and lines 389-395 except those parts that concern the differences in centralized versus decentralized test distribution models.

**Do you want your identity to be public for this peer review?** For information about this choice, including consent withdrawal, please see our Privacy Policy

Reviewer #1: No

Reviewer #2: No

Reviewer #3: No

---

## [Decision Letter · Decision Letter 1]

8 Sep 2025

PGPH-D-25-00883R1

Acceptability of COVID-19 Self-testing among Social and Clinical Vulnerable Populations using a Decentralized Testing Model in Abuja, Nigeria; A mixed methods analysis of an implementation study.

Dear Dr. Dunkley,

Thank you for submitting your manuscript to PLOS Global Public Health. After careful consideration, we feel that a few minor edits would enable the manuscript to fully meet PLOS Global Public Health’s publication criteria. Therefore, we invite you to submit a revised version of the manuscript that addresses the points raised during the review process.

We look forward to receiving your revised manuscript.

Kind regards,

Shifa S. Habib

Academic Editor

Journal Requirements:

Reviewers' comments:

Reviewer's Responses to Questions

**Comments to the Author**

Reviewer #1: All comments have been addressed

Reviewer #3: All comments have been addressed

publication criteria?

Reviewer #1: Yes

Reviewer #3: Yes

3. Has the statistical analysis been performed appropriately and rigorously?

Reviewer #1: Yes

Reviewer #3: Yes

4. Have the authors made all data underlying the findings in their manuscript fully available (please refer to the Data Availability Statement at the start of the manuscript PDF file)?

Reviewer #1: Yes

Reviewer #3: Yes

5. Is the manuscript presented in an intelligible fashion and written in standard English?

Reviewer #1: Yes

Reviewer #3: Yes

Reviewer #1: The authors have addressed all my previous comments and I do not have any further comments.

Reviewer #3: I would like to thank their authors for the thorough implementation of the reviewers’ comments. Besides four very minor suggestions (see below), I don’t have any further comments.

- Please add the link (https://datacompass.lshtm.ac.uk.) to the online data repository to the main manuscript. This could either be done through a ‘Data availability’ statement at the end of the manuscript, or by including mentioning it in the methods section.

- Line 78 to 81: Please revisit this sentence, as it seems to be missing words.

- Line 288: I would suggest to add a comma after ‘populations’. Please also check the remaining manuscript for any potential typographical inconsistencies.

- Line 318: Please complete the section 'Author contributions'

**Do you want your identity to be public for this peer review?** For information about this choice, including consent withdrawal, please see our Privacy Policy

Reviewer #1: No

Reviewer #3: No

---

## [Decision Letter · Decision Letter 2]

4 Dec 2025

Acceptability of COVID-19 Self-testing among Social and Clinical Vulnerable Populations using a Decentralized Testing Model in Abuja, Nigeria; A mixed methods analysis of an implementation study.

PGPH-D-25-00883R2

Dear Ms Dunkley,

We are pleased to inform you that your manuscript 'Acceptability of COVID-19 Self-testing among Social and Clinical Vulnerable Populations using a Decentralized Testing Model in Abuja, Nigeria; A mixed methods analysis of an implementation study.' has been provisionally accepted for publication in PLOS Global Public Health.

Best regards,

Julia Robinson

Executive Editor

Reviewer Comments (if any, and for reference):

Reviewer's Responses to Questions

**Comments to the Author**

Reviewer #1: All comments have been addressed

Reviewer #3: All comments have been addressed

publication criteria?

Reviewer #1: Yes

Reviewer #3: Yes

3. Has the statistical analysis been performed appropriately and rigorously?

Reviewer #1: Yes

Reviewer #3: Yes

4. Have the authors made all data underlying the findings in their manuscript fully available (please refer to the Data Availability Statement at the start of the manuscript PDF file)?

Reviewer #1: Yes

Reviewer #3: Yes

5. Is the manuscript presented in an intelligible fashion and written in standard English?

Reviewer #1: Yes

Reviewer #3: Yes

Reviewer #1: The authors have addressed all my prior comments and I have no additional comments.

Reviewer #3: Thank you very much for addressing the remaining comments! I have no further suggestions.

**Do you want your identity to be public for this peer review?** For information about this choice, including consent withdrawal, please see our Privacy Policy

Reviewer #1: No

Reviewer #3: No
